# Targeting anger for COVID-19 prevention: The motivating role of anger on media use and vaccination intention

Yoo Jung Oh[1]*, Muhammad Ehab Rasul[2], Jong In Lim[1], Christopher Calabrese[3], Emily McKinley[4], Hannah Stevens[5], Monique Mitchell Turner[1], Maria Knight Lapinski[1], Tai-Quan Peng[1]

1 Department of Communication, Michigan State University, East Lansing, Michigan, United States of America, 2 National University of Singapore, Centre for Trusted Internet and Community, Singapore, Singapore, 3 Department of Communication, Clemson University, Clemson, South Carolina, United States of America, 4 Department of Communication, University of California, Davis, Davis, California, United States of America, 5 Department of Communication, University of California, Santa Barbara, Santa Barbara, California, United States of America

* ohyoojun@msu.edu

## Abstract

The COVID-19 pandemic resulted in public anger due to its disruptive and harmful nature. However, anger remains an understudied concept despite its potential to persuade the public and spark action. The current study investigates the role of anger in driving COVID-19 vaccination intentions. In Study 1, through a rolling-cross sectional survey of U.S. adults during the early stages of the COVID-19 epidemic ($N=6,141$), it was found that anger towards COVID-19 was associated with increased use of social and traditional media news, which was then related to improved vaccination intentions. In Study 2, utilizing computational analysis of a Twitter (now X) dataset using an AI classifier, 15 targets of anger were identified from real-world anger expressions in social media discourse about COVID-19. Building on these insights, Study 3 involved a representative survey of U.S. adults during the post-emergency declaration stage of COVID-19 ($N=1,005$). This survey aimed to replicate the findings of Study 1 while incorporating the anger targets identified in Study 2. The results revealed that different targets of anger were associated with vaccination intentions through the consumption of traditional news media. Although social media was a prominent channel for news about vaccination at the beginning of the pandemic, our findings suggest that traditional media news use may be an important link in understanding the relationship between anger and vaccination intentions. Theoretical, practical, and methodological implications are discussed.

**Data availability statement:** The coded datasets are available at the following publicly accessible OSF link: https://osf.io/6nyhz/.

**Funding:** Study 1 was funded by a RAPID Grant from the National Science Foundation (award number, 2029633; PI Monique M. Turner) The funders did not play any role in the study design, data collection and analysis, decision to publish, or preparation of the manuscript.

**Competing interests:** The authors have declared that no competing interests exist.

## Introduction

Since 2019, individuals across the globe have experienced severe effects of the novel coronavirus (COVID-19). In the early stages of the pandemic, public health officials and politicians were forced to issue stay-at-home orders and vaccine mandates [1]. These mandates co-existed with anger and frustration among political elites and the public, with some arguing that the mandates violated personal freedoms [2,3]. Studies documented that the onset of COVID-19 was related to anxiety and mental stress world-wide as the threat of the pandemic grew [4,5]. Theorists of emotion have shown that an event like COVID-19 can induce various emotions depending on how the individual appraises it [6,7]. Particularly in the context of COVID-19, anger was one of the core emotional experiences individuals reported regarding the pandemic [8]. More specifically, travel bans, vaccine mandates, and physical distancing measures triggered anger as a salient emotional response to them [9].

Anger during the COVID-19 pandemic played a critical role in shaping vaccine-related attitudes and behaviors. Research shows that anger is associated with heightened anti-vaccine attitudes (e.g., [10]) while also serving as a motivator for addressing misinformation (e.g., [11]). On one hand, anger can promote adaptive actions, such as supporting public health measures. On the other hand, it can deepen polarization or fuel resistance to change. This dual nature of anger highlights its potential to be both a barrier and a facilitator of public health initiatives.

Although there is research focused on anger and COVID-19 (e.g., [11,12]), and social media use and COVID-19 vaccine hesitancy [13,14], less attention has been paid to investigating how anger and media use collectively are associated with COVID-19 vaccine behavior despite the need for improving vaccine uptake. Prior work has found that refutational messages may influence general vaccination attitudes through reduced anger [15], and that anger towards COVID-19 is associated with COVID-19 media exposure, but not general vaccine attitudes [16]. However, less is known about anger as a potential driving force for news media use, as well as for COVID-19 vaccination intentions, since intentions are a more direct, specific predictor of behavior than general attitudes [17,18].

To address this gap, across three studies, we investigated how anger and news consumption, including both social and traditional media news use, are associated with COVID-19 vaccination behavioral outcomes. Further, we examine how various targets of anger manifest in public discourse on social media. Specifically, in Study 1, we explore whether anger is associated with intention to get the COVID-19 vaccine and whether this relationship is mediated by social media and traditional media news use. Then, in Study 2, using observational social media data, we focus on how anger manifests in public discourse about COVID-19 and vaccines, identifying the key targets of anger within this context. Ultimately, in Study 3, using a representative sample of U.S. citizens, we replicate and extend the findings of Study 1 by exploring the identified targets of anger and their associations with news consumption and future vaccination intentions. Extending previous research, we argue that the driving force of anger on vaccination behavior is dependent on the *who* the blamable target of anger is. Across three studies at the early, middle, and post-emergency declaration

stages of the COVID-19 pandemic, we aim to capture who and what people are angry toward and the implications of emotion on news media consumption and behavioral intentions to vaccinate.

## COVID-19-induced anger and behavioral outcomes

Anger, ranging from irritation to fury, involves its unique appraisal patterns (i.e., third party responsibility, high certainty, and control [7]) and tendency to provoke actions [19]. In theory, what distinguishes anger from other emotions is its inherent blame attribution. Anger involves a blamable target who is perceived to be responsible for obstructing one's personal goals [7,19,20]. Additionally, anger is considered an approach emotion (i.e., one that drives behavior, rather than an avoidance mechanism) because angry individuals are more prone to take action to cope with their obstructed goals [21,22]. Anger is uniquely characterized by its tendency to attribute responsibility to specific agents and motivate responses to those blamed targets. Due to this unique nature, anger's potential to lead to persuasive outcomes has received attention. The cognitive functional model [23] suggests that, in certain conditions, anger can lead to motivated attention and motivated information processing, which in turn may influence message acceptance. Also, the anger activism model [24,25] posits that anger interacts with efficacy beliefs to elicit intentions to engage in prosocial behaviors.

Anger has been found to be one of the most prevalent forms of emotional engagement with COVID-19-related issues (e.g., [8,26]). Studies have documented anger was directed towards various targets including individuals [27], the government (e.g., [28]), and related policies (e.g., prevention mandates, such as stay-at-home orders; [29]) during the COVID-19 pandemic as these policies were perceived as an attack on personal freedoms [2,3]. This can be explained by psychological reactance theory, which suggests perceived threats to one's autonomy or freedom through messages can result in resistance, anger, and hostility [30]. Indeed, research focused on COVID-19 has found that psychological reactance resulted in anger and resistance from anti-mask individuals [31]. The extensive COVID-19-related restrictions, including stay-at-home orders, likely intensified public anger due to the unprecedented scale of government-mandated behavior change. Based on the intertwined-process cognitive-affective model, which suggests that reactance can be cognitive (and affective negative thoughts and anger, see [32]) and the magnitude of request for behavioral change can significantly increase reactance [33].

During the pandemic, anger was associated with health-related behavioral outcomes such as vaccine hesitancy (e.g., [34,35]). People who were angry at vaccination mandate policies expressed their anger and frustration by actively refusing to receive COVID-19 vaccines (e.g., [36]). This anger fueled subsequent vaccine hesitancy, which worsened the pandemic's harmful effects and likely led to increased death tolls [37]. However, depending on the target of anger, anger can elicit prosocial health behavioral responses. For instance, one study found that anger directed at individuals violating public health recommendations led to greater support for stricter COVID-19 policies [38]. This demonstrates that identifying the target of anger is essential to understanding its consequences for public health. The specific properties of anger as an emotion prompt further investigation of its association with vaccination behavior in the context of COVID-19. Thus, we propose the following research question:

**RQ1**: How is COVID-19 targeted anger associated with vaccination intention?

## Media use as the mediator

The process through which anger may specifically impact COVID-19 vaccination behavioral intention bears further examination. One explanation is that the action tendency of anger may prompt individuals to seek out information to remedy or defend oneself from a threat [23,39]. For example, recent work found that as an approach-oriented emotion, anger led to increased message processing times and information seeking consistent with its behavioral tendency (i.e., seeking information related to attribution rather than prevention) [40]. When people experience anger about health threats such as COVID-19, they are likely to engage in news media consumption both to identify who or what is to blame for the threat

and to seek guidance [41]. Prior research has shown that times of crisis and uncertainty are accompanied by increases in news media consumption [42,43]. However, not all media serve the same role. Traditional media outlets often provide live briefings from public health officials and leaders, positioning them as primary sources of information [44]. At the same time, in times of crisis, the preference for immediate news [43] makes social media an attractive source of information. During the COVID-19 pandemic, an unprecedented demand for information, combined with stay-at-home orders and an increase in remote work resulted in high reliance on social media for information [45]. Taken together, these patterns suggest that it is essential to understand not only whether anger drives information seeking, but also the types of media individuals turn to when experiencing anger toward COVID-19 [46–48]. Notably, two prominent types of information channels used for COVID-19 are social and traditional news media [49,50].

## Anger and social media news use

During the COVID-19 pandemic, people experienced heightened emotions such as stress, anxiety, and anger as they encountered novel changes in their daily lives, a phenomenon scholars have termed *pandemic rage* [51]. To cope with the significant disruptions caused by stay-at-home orders, lockdowns, and physical distancing, people increasingly turned to online communities, particularly social media platforms such as Facebook and Instagram. During COVID-19, social media served as a channel of easy access to information [52], as well as an outlet for people to ease the stress and anxiety caused by the pandemic by sharing stories of hardship and engaging with others in a mediated environment [53,54]. Importantly, social media also became a platform for the public to express anger towards economic, social, and COVID-19-related issues [55].

## Anger and traditional media news use

In addition to social media, traditional media outlets such as television and newspapers served as critical information channels during COVID-19. According to a Pew survey, 53% of Americans reported relying on traditional media news for information about COVID-19 [56]. Existing research has argued that negative emotions such as anger can prompt heightened engagement with information due to increased arousal [57,58]. Anger has been shown to be related to increased information seeking from newspapers and magazines, in addition to social media platforms [59]. The anger people experienced during COVID-19 could have motivated them to seek information for various reasons, such as to assign blame or validate their beliefs [58]. In addition, when high anger is coupled with high efficacy, it can motivate individuals to seek information and advocate for vaccine policy changes [60].

## Consequences of media use

Several researchers have demonstrated that social media news use is related to COVID-19 vaccine hesitancy [14,46,61,62]; but the findings remain mixed. Higher levels of social media use is related to lower compliance with public health protocols related to COVID-19 [63]. Yet, other research has found social media use was positively associated with COVID-19 vaccination across the political spectrum [46]. Further, existing research on traditional media's impact on COVID-19 behaviors also shows mixed results. Some studies found that increased traditional media news consumption was associated with lower compliance with COVID-19 guidelines, particularly among Fox News viewers [64]. Conversely, others have found that traditional media news use is associated with increased compliance with public health protocols to prevent COVID-19 [50,65]. While news media use and vaccination behavioral outcomes have been shown to be related, it is still unclear whether news media use has a positive or negative association to vaccination outcomes. Thus, we proposed the following research question:

**RQ2**: Do (a) social media and (b) traditional media news use mediate the relationship between COVID-19 targeted anger and vaccination behavioral outcomes?



## Study 1

In Study 1, we explored the role of anger towards COVID-19 as a potential motivator for vaccination intention. Furthermore, we investigated how news media consumption may serve as a mediator in this relationship.

### Study 1 Method

**Sampling design and sample description.** We recruited participants using a non-probability, quota-based sampling approach across 20 states in the U.S., specifically chosen to capture a range of COVID-19 prevalence rates at the time of the survey (July 6 – October 16, 2020). The sampling strategy included grouping states by COVID-19 prevalence and applying quotas for demographics like age, sex, race, and education to ensure representation. Data collection took place in 17 waves (500 participants per wave), with oversampling to meet quotas, and the final sample demographics aligned closely with the target population. After removing outliers ($n = 88$) and missing data (n = 2,549) from the initial 8,778 participants, our final analysis included 6,141 responses.

The average age of participants was 42.80 years ($SD = 16.79$, ranging 18–91 years old). Slightly over half of participants identified themselves as biological females (53.1%, n = 3,261). The racial composition was meant to look similar to the US population with an over-sample of Black/African Americans: White/Caucasian (62.6%), Black/African American (17.3%) and Asian (6.9%). Most participants reported having a college-level education or higher (69.1%). 43.3% of the sample reported that they held a liberal political ideology, while 56.7% of the participants indicated they were conservative.

Protocols were reviewed and approved by one of the author's university Institutional Review Board (IRB number: 00004287). Informed written consent was obtained via a "consent" section at the beginning of the online survey questionnaire, allowing voluntary participation. All research was carried out in accordance with relevant guidelines and regulations.

**Measures.** To measure **anger towards COVID-19**, a single-item thermometer scale was used. Participants rated their anger from 0 (none) to 100 (overwhelming) in response to the statement: "How much of the following emotions would you say you are experiencing when it comes to COVID-19?". This scale quantified the emotional impact of COVID-19 on individuals. **Social media news use** was measured by how frequently individuals accessed COVID-19 news via social media platforms like Facebook, Twitter, or Instagram using a 5-point Likert-type scale (1 = Almost all the time, 5 = Never; reverse-coded such that higher values represent greater frequency). To measure **traditional media news use**, participants reported the number of minutes they spent on local or national news across all mediums each day (TV, internet, newspapers, etc). Preliminary data analysis revealed that the raw media consumption data exhibited an extremely wide range, spanning from 0 minutes to one quadrillion minutes. Considering this extensive variability, we constrained the data to a range of 0–480 minutes (8 hours), excluding 88 responses that exceeded this threshold from the main analysis. Lastly, to measure **COVID-19 vaccination intention,** a single-item thermometer scale (ranging from 0 to 100) was used. Participants rated their willingness to receive COVID-19 vaccines in response to the statement: "If a vaccine for COVID-19 became available, how likely would you be to receive the vaccination within 6 months of its availability?" Demographic variables such as age, sex, race, education level, and political ideology were also assessed. See Table 1 for descriptive statistics and correlations among study variables.

### Study 1 Results

A parallel mediation model was adopted (PROCESS MACRO Model 4, [66]) to examine the mediating role of both social media news use and traditional media news use on the relationship between anger towards COVID-19 and vaccination intention (Fig 1). Age, sex, race, education level, and political ideology were included in the model as covariates. The total effect model indicated that anger towards COVID-19 was positively related to COVID-19 vaccination intention, $b = 0.033$, $SE = 0.013$, $t = 2.46$, $p = 0.014$. The total effect model explained 9.3% of the variance in vaccination intention $R^2 = 0.093$, $F_{(9, 6131)} = 69.810$, $p < .001$. The direct effect of anger on vaccination intention was not significant, $b = 0.021$, $SE = .013$,

**Table 1. Zero Order Correlation between Key Variables with Descriptive Statistics (N = 6,141).**

| | M | SD | 1 | 2 | 3 | 4 |
|---|---|---|---|---|---|---|
| 1. Anger towards COVID-19[a] | 52.50 | 34.03 | – | | | |
| 2. Social media news use[b] | 3.21 | 1.28 | .09** | – | | |
| 3. Traditional media news use[c] | 66.31 | 68.73 | .07** | .03* | – | |
| 4. COVID-19 vaccination intention[a] | 55.43 | 36.97 | .01 | −.001 | .12* | – |

Note. [a]Scale ranged from 1 to 100; [b]Scale ranged from 1 to 5; [c]Number of minutes.

* $p < .05$, ** $p < .01$.

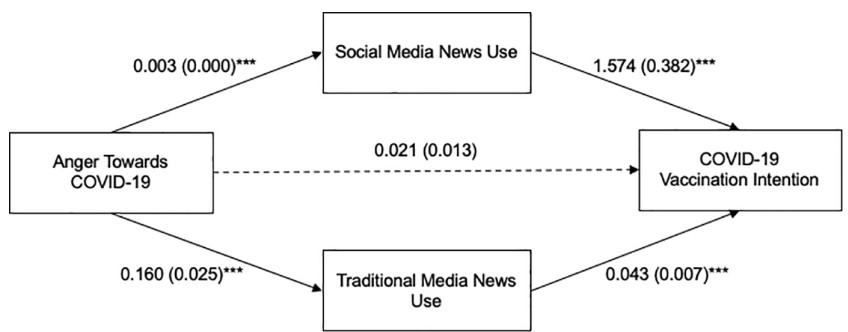

**Fig 1. Parallel Mediation Model of the Relationship between Anger and COVID-19 Vaccination Intention through Social Media and Traditional Media News Use.**

$t = 1.565$, $p = .118$. Anger towards COVID-19 was positively related to social media news use, $b = 0.003$, $SE = .000$, $t = 7.147$, $p < .001$, as well as traditional media news use, $b = 0.160$, $SE = .025$, $t = 6.347$, $p < .001$. Greater social media news use was associated with higher vaccination intention, $b = 1.574$, $SE = .382$, $t = 4.124$, $p < .001$. Similarly, greater traditional news use was associated with higher vaccination intention, $b = 0.043$, $SE = .007$, $t = 6.336$, $p < .001$.

Note. Unstandardized B was used to indicate path coefficients, with standard errors in parentheses. Paths represent associations, not causal relationships. Age, sex, race, education level, and political ideology were included in the model as covariates. ***$p < .001$.

The indirect effect of anger on vaccination intention via social media news use was significant with a point estimate of .005 (Boot $SE = .001$, 95% CI = [.003,.008]), and the indirect effect via traditional news use was also significant with a point estimate of .007 (Boot $SE = .002$, 95% CI = [.004,.010]), indicating partial mediation of social media news use and traditional news use on the relationship between anger and vaccination intention. Including social media and traditional media news use as mediators in the model significantly increased the explained variance in future vaccination intention ($R^2 = 0.102$, $\Delta R^2 = 0.009$, $p < .001$).

## Study 1 Discussion

Findings from Study 1 indicate that during the initial stages of the COVID-19 pandemic, prior to the first nationwide administration of the COVID-19 vaccine, anger towards COVID-19 was associated with increased vaccination intentions among participants. Further, mediation analysis revealed that the relationship between anger and vaccination intentions was associated with increased media use, such that anger towards COVID-19 was related to increased social media and traditional media news use, which in turn motivated individuals to receive the vaccine.

Our findings align with existing research which suggests that anger as a discrete emotion may drive action to defend oneself [7]. That is, people who were angry, especially during the beginning of the pandemic, may have been prompted to seek information through news media to protect themselves from the disease. During this process, traditional media may have served a key role in disseminating health messages, providing expert insights, and emphasizing the impact of COVID-19 [50]. On the other hand, social media may have served as a dynamic platform, offering real-time updates, enabling the sharing of personal experiences, and facilitating lively discussions about personal experiences and perspectives related to the disease [67]. As shown in our findings, it is likely that both traditional news media and social media news significantly contributed to COVID-19 vaccination behavior by serving as means of gathering and utilizing the information and forming a positive perception about vaccination.

What remains unclear is whether COVID-19, the virus itself, was the sole focus of individuals' anger during the pandemic. Theoretically, anger often rises towards a blamable target [7,19]. Given that anger towards COVID-19 can include various blamable targets such as government authorities, healthcare systems, and even individuals who do not follow safety guidelines, it is important to consider these varied factors that may have led to media use and vaccination intention outcomes.

Thus, a necessary precursor to understanding the theoretical and practical implications of anger involves a detailed identification of the key targets of anger associated with COVID-19. As such, in Study 2, we utilize a large-scale, unobtrusive behavioral dataset that involves individuals' expression of anger about COVID-19 on social media. Through a content analysis of a weekly representative set of tweets surrounding COVID-19 across a 12-month period, we provide a more nuanced understanding of anger targeted towards various COVID-19 entities.

## Study 2

Study 1 laid the groundwork by demonstrating that anger directed at COVID-19 was positively associated with vaccination intentions through increased activity on social media and traditional news media use. However, anger is a multifaceted emotion with varying targets and expressions [7,19]. That is, anger's influence on vaccination behaviors can vary by what or at whom individuals are angry [68]. This more nuanced understanding of anger is crucial for devising effective strategies to promote vaccine acceptance and mitigate public health risks.

To further investigate the nuanced role of anger on vaccination intention in COVID-19 discourse, Study 2 utilizes a dataset of tweets discussing COVID-19 vaccines from September 14, 2020, to October 1, 2021, to explore anger's manifestations in the context of the COVID-19 pandemic. This time period was selected to include three months before the first COVID-19 vaccination administration on December 14, 2020 [69], providing a full overview of individuals' perceptions surrounding the COVID-19 vaccine during this crucial period. The main objective of Study 2 is to discern the various forms of COVID-19-induced anger by analyzing tweets related to anger and COVID-19, with a predominant focus on vaccine-related discussions. Through both manual and AI-assisted coding, we identified 15 different targets of anger related to COVID-19.

### Study 2 Method

**Twitter Dataset.** Given our focus on vaccination behavior, we utilized Twitter/X's API to scrape tweets containing keywords related to the COVID-19 vaccine from September 14, 2020, to October 1st, 2021. Keywords included "Moderna", "Pfizer", "Johnson", "Dose", "Vax", "Vaccine", "Vaccines", "Vaccinate", "Vaccinated", and "Vaccinating". A systematic random sample of 7,000 tweets per week was selected to obtain a representative weekly sample of tweets; the sample contained 375,000 tweets. Then, we employed a dictionary-based text analysis tool (LIWC-22 [70]), to examine the emotions in each tweet. We identified 5,730 tweets (1.55%) that contained anger-related words, encompassing our full dataset for analysis.

**Targets of Anger Classification.** We analyzed a dataset of 5,730 anger-related tweets to identify specific targets of anger. Given that several targets of anger were identified in previous studies such as anger towards individuals who

do not adhere to safety guidelines or are reluctant to receive vaccination, those who had vaccine opportunity ahead others (i.e., vaccine envy), government, and policies regarding COVID-19 (e.g., [27–29,71]), we developed a preliminary codebook informed by prior literature. Then, in the manual review process, two researchers independently examined subsets of tweets to assess whether the preliminary categories adequately captured the expressions of anger, while also noting any emergent targets not reflected in prior work. Through iterative discussion, we refined the codebook by merging overlapping categories, clarifying definitions, and adding new categories that were inductively derived from the data.

Human coding was applied to 20% of the total dataset, which consisted of 1200 tweets. From this subset, a first random batch of 240 tweets was independently coded by two researchers. We used Krippendorf's alpha to measure and establish an acceptable level of intercoder reliability. Following discussions and clarifications to the codebook, a randomly selected batch of 240 tweets was coded. We achieved an intercoder reliability with a Krippendorff's alpha of 0.725, which met the acceptable standards [72]. Any discrepancies were resolved, and each coder independently coded the remaining tweets.

For automated coding of targets of anger, we selected the GPT model for its proven efficiency and accuracy, as demonstrated in previous validation studies [73,74]. Specifically, we fine-tuned the GPT-3.5 model [75] through *instruction tuning* [76], which included detailed guidelines and a codebook (see S1 File) to instruct the model on how to classify the tweets and through *supervised fine-tuning* [77] by providing a training dataset with human-coded outputs. We used 90% of the human-coded dataset as the training set, while the remaining 10% served as the test set to evaluate the model's performance. The model's performance on the test set was evaluated, yielding an accuracy of 71.67%, a precision of 73.30%, and recall of 71.67%, and an F1 score of 70.98%. These results indicated that the model is balanced and acceptable for our classification task [78]. Following this validation, the fine-tuned model was employed to code the remainder (4,530 tweets; 80%) of the dataset.

## Study 2 Results

Table 2 presents a comparison of targets of anger related to COVID-19 and their corresponding frequencies and percentages in two datasets: Human (20% of the dataset) and Fine-tuned GPT-3.5 coded (80% of the dataset). There was a notable expression of anger towards anti-vaxxers or vaccine-hesitant individuals (10.0% Human; 9.42% GPT). Additionally, public health officials were also a significant target of public anger (8.50% Human; 6.86% GPT). Other targets of anger included the handling of COVID-19 by various authorities: government, governor, politicians or political parties (7.67% Human; 7.39% GPT), President Trump (5.17% Human; 6.47% GPT), and medical institutions or healthcare system (3.67% Human; 2.89% GPT). Anger towards vaccine and vaccine mandate (5.17% Human; 5.76% GPT), misinformation spreaders (3.75% Human; 5.14% GPT), and vaccine envy (4.50% Human; 4.02% GPT) were also expressed in tweets. Lastly, findings show minimal mentions of targets such as the World Health Organization's handling of COVID-19 (0% Human; 0.20% GPT), pharmaceutical companies (0.83% Human; 0.53% GPT), masking and social distancing (0.83% Human; 0.49% GPT), individuals who do not adhere to masking and social distancing guidelines (1.17% Human; 1.04% GPT), and former President Biden (0.83% Human; 0.49% GPT). One category, "anger towards employers mandating COVID-19 vaccination," was removed from analysis due to only a single case across the dataset, making it analytically uninformative. In contrast, although the category "anger towards World Health Organization's handling of COVID-19" did not appear in the human-coded dataset, we retained it because multiple instances emerged in the GPT-coded portion, indicating that they reflected a recurring theme in the larger dataset.

Notably, approximately 48% of the tweets in the dataset were placed in the 'Others (unspecified)' category, indicating they did not clearly fall into any of the predefined anger categories. Given the possibility that the initial LIWC-based screening may have incorrectly flagged some non-anger tweets as containing anger, and that the 'Others' category could also include emergent expressions of anger not captured by our predefined categories, we conducted an additional validation analysis. Two researchers independently coded a random sample of 200 tweets from this category for the presence

**Table 2. Comparison of Targets of Anger related to COVID-19 in Human and Fine-Tuned GPT-Coded Datasets.**

| | Human | | Fine-tuned GPT3.5 | |
|---|---|---|---|---|
| | Count | % | Count | % |
| 1. COVID-19 virus and its negative social and health effects on people | 10 | 0.83 | 40 | 0.88 |
| 2. Vaccine and vaccine mandate | 62 | 5.17 | 261 | 5.76 |
| 3. Personal dislike of wearing masks and social distancing | 10 | 0.83 | 22 | 0.49 |
| 4. Anger towards those not adhering to mask-wearing and social distancing guidelines | 14 | 1.17 | 47 | 1.04 |
| 5. President Trump's handling of COVID-19 | 62 | 5.17 | 293 | 6.47 |
| 6. Former President Biden's handling of COVID-19 | 10 | 0.83 | 43 | 0.95 |
| 7. Government, governor, politicians or political parties handling COVID-19 | 92 | 7.67 | 335 | 7.39 |
| 8. Jealousy or envy experienced when others are given the opportunity to receive a COVID-19 vaccine | 54 | 4.50 | 182 | 4.02 |
| 9. Anti-vaxxers or vaccine-hesitant individuals for COVID-19 | 120 | 10.0 | 427 | 9.42 |
| 10. Misinformation spreaders of COVID-19 | 45 | 3.75 | 233 | 5.14 |
| 11. Public health officials during COVID-19 | 102 | 8.50 | 311 | 6.86 |
| 12. Pharmaceutical companies handling COVID-19 | 10 | 0.83 | 24 | 0.53 |
| 13. Medical institutions or Healthcare system | 44 | 3.67 | 131 | 2.89 |
| 14. World health organization handling COVID-19 | 0 | 0 | 9 | 0.20 |
| 15. Others (unspecified) | 565 | 47.08 | 2172 | 47.94 |
| Total | 1200 | 100% | 4529 | 99.98% |

*Note.* One category, "anger towards employers mandating COVID-19 vaccination," was deleted due to significantly low occurrence (n = 1; 0.02% from GPT-coded dataset).

or absence of anger, achieving strong intercoder reliability (92% agreement, Krippendorff's α = 0.79). We then asked the GPT model to code the same 200 tweets for anger presence. Comparing GPT's classifications with human coding showed strong performance (Krippendorff's α = 0.82, Accuracy = 93%, Precision = 0.786, Recall = 0.957, F1-score = 0.863). Following this validation, we applied the model to classify all 2,172 tweets in the 'Others (unspecified)' category for anger presence (See S2 File for prompts).

Our subsequent validation analysis revealed that this category contained 803 tweets (36.97%) that expressed anger without clear targets and 1,369 (63.03%) that did not express anger. To further investigate the nature of tweets categorized as "Others", we qualitatively reviewed examples from both subcategories.

Among tweets expressing anger without specific targets in the "Others" category, many concerned unrelated issues or directed anger at vague or generalized targets. For example, one user expressed frustration over the annual need to get a flu shot, highlighting the stress it caused, yet not directly tying it to COVID-19 ("Every year this season, I'm so crazy stressful because I need to get the flu shot vaccine, I hate it, but I need to take it. Now, I'm done... I took it today and my stress is released. #FluShot #FluShot2020"). Another example displayed anger directed at vague or generalized targets (e.g., "It's really irking me that decent people wished him well. He only cares about himself and the money he will cash in on the new vaccines. We will never know how many could have been saved were it not for his mishandling").

The non-anger tweets primarily fell into two types: those sharing information without emotional content (e.g., "Went to same store my son & I were mocked for wearing masks a few months ago, a nice store in nice area. Happy to say this time 75% of the many shoppers were masked & no one said anything rude") and those suggesting pragmatic solutions to the pandemic (e.g., "Achieving population immunity which means the spread is low and stable not necessarily that everyone is literally immune is the only real solution. The argument is how to achieve it with the lowest cost in terms of mortality and public health"). These non-anger tweets likely resulted from LIWC's initial screening incorrectly flagging certain language patterns as anger-related.

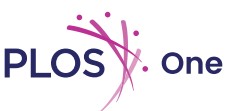

**Study 2 Discussion**

This study primarily focused on expressed anger related to COVID-19 on Twitter (now X) over a year period, including when the COVID-19 vaccine was first available. Using a fine-tuned GPT-3.5 model and a comprehensive codebook, we identified 15 different targets of anger that manifested on social media. Notably, classification results suggest that the most frequent identified target of anger was directed at people opposing vaccines, such as anti-vaxxers and vaccine-hesitant individuals, likely due to frustration with perceived threats to public health and safety. Moreover, significant anger towards public health officials and political leaders indicates dissatisfaction with the management of the pandemic and the politicization of health measures.

The use of the fine-tuned GPT-3.5 model significantly advanced our analysis of large-scale social media data. This AI-driven approach, as validated in previous studies (e.g., [73,74]) can enhance efficiency and consistency in handling complex datasets. By incorporating prompt-engineering techniques, it facilitated independent verification of results and increases the replicability of research findings. Building on our systematically identified targets of anger, the next crucial step is to replicate and extend our findings from Study 1. This involves examining how anger directed at different targets is associated with news media consumption, and vaccination behavioral outcomes.

## Study 3

While Study 1 found that both social media and traditional media news use mediated the relationship between general anger towards COVID-19 and vaccination intentions prior to the first administration of the vaccine, Study 2 identified 15 different targets of anger associated with COVID-19 on social media before and after COVID-19 vaccines were available. What remains unclear is whether anger towards each specific target is similarly associated with news media consumption in the post-emergency declaration era of COVID-19, as well as on vaccination intentions for future pandemics.

Now past the initial stages of the pandemic, uncertainty and anxiety over COVID-19 has diminished, where only 20% of U.S. adults perceive COVID-19 to be a public health threat [79]. Thus, while all traditional media viewership has declined in recent years [80], individuals may have turned back to traditional outlets for news, in addition to social media outlets. Similarly, with the politicization of COVID-19 over time [81], social media may no longer be a primary channel for acquiring health information.

Furthermore, since the specific targets of anger vary, we propose that the relationship between anger and media consumption can vary by the particular target of the anger. For instance, individuals who express anger towards anti-vaxxers might prefer traditional media because these outlets provide more formal discussions and reports on governmental policies. Conversely, those who are angry at vaccine mandates might be more inclined to seek out social media, where they can engage directly with opposing views and find community support.

Through a survey of U.S. adults, we examined the relationships between anger towards specific targets, media consumption, and future vaccination behavior. We first categorized the main targets of anger through exploratory factor analysis. Then, we examined whether each specific target of anger was associated with news media use and vaccination intentions. Lastly, we ran mediation path analyses to explore potential mechanisms between anger and future vaccination intentions.

### Study 3 Method

We conducted an online survey (*N* = 1,005) of U.S. adults through Prolific ([www.prolific.com](www.prolific.com)) during the period of April 1–18, 2024. The sample matched the U.S. census in terms of three key demographic characteristics, age, sex, and ethnicity. The average age of participants was 47.41 years (*SD* = 17.22, ranging 18–95 years old). Slightly over half identified themselves as biological females (50.1%, n = 487). The racial composition was predominantly White/Caucasian (63.5%), followed by Black/African American (11.8%), Hispanic/Latino (9.6%), Multiracial (6.6%), and Asian (5.7%). Approximately 71% of the participants had attended some college or completed a bachelor's degree. Regarding political ideology, 67.4% of our sample identified as being liberal while 32.6% identified as being conservative.



Protocols were reviewed and approved by one of the author's university Institutional Review Board (IRB number: 00010493). Informed written consent was obtained via a "consent" section at the beginning of the online survey questionnaire, allowing voluntary participation. All research was carried out in accordance with relevant guidelines and regulations.

## Measures

Based on prior classification of COVID-19-related social media posts in Study 2, we created a set of feeling thermometer items ranging from 0 to 100 to estimate **anger towards specific targets** related to COVID-19 (e.g., "When I think about vaccines and vaccine mandates, I feel angry"). It should be noted that in Study 1, media news use variables were measured using different scales, which limited the comparability of these variables. To address this, in Study 3, we measured media news use variables using similar scales to improve comparability. Specifically, **traditional media news use** was measured using a composite score for television and newspaper use; **social media news use** was measured using a composite score for Facebook, Twitter, Instagram, and TikTok, where each item ranged from 1 (never) to 7 (6–7 days). Lastly, we assessed **vaccination intentions** on a scale from 1 (very unlikely) to 5 (very likely) using the following: "In the event of a future pandemic similar to COVID-19, and a vaccine becomes available, how likely are you to get vaccinated within the first 6 months of its availability?".

## Study 3 Results

The study first employed an Exploratory Factor Analysis (EFA) with Principal Component Analysis (PCA) as the extraction method to explore the underlying factor structure of the 15 types of anger regarding COVID-19. Prior to the analysis the suitability of the data of factor analysis was assessed. The Kaiser-Meyer-Olkin (KMO) measure of sampling adequacy returned a value of .91, indicating the sample was suitable for EFA. Bartlett's Test of Sphericity was significant, $\chi2(105) = 9370.1$, $p < .001$, suggesting that the items were sufficiently correlated for PCA.

Initially, the number of components to be extracted was determined using Kaiser's criterion (eigenvalues greater than 1), which resulted in a three-factor solution. However, after investigating the factor loadings, one item (general anger towards COVID-19) indicated a low loading score across the components (<.50). Thus, it was concluded that a four-factor solution was deemed most interpretable to represent the different aspects of anger. Accordingly, we extracted four principal components, which explained 73.66% of the total variance in the data.

Anger Factor 1 accounted for 41.01% of the variance, and was characterized by items related to anger expressed towards **health officials and agencies** (public health officials, World Health Organization, pharmaceutical companies, medical institutions or healthcare system), career **politicians** (politicians or political parties, former President Biden, governors and their policies), and **COVID-19 preventive health behaviors** (vaccines and vaccine mandates, mask-wearing and social distancing). A follow-up reliability analysis found factor 1 to have excellent internal consistency ($\alpha = .92$).

Anger Factor 2 explained 20.23% of the variance, with items related to anger expressed towards **anti-vaxxers or vaccine-hesitant individuals**, **individuals not adhering to mask-wearing and social distancing guidelines**, **misinformation spreaders**, and **President Trump**. It is important to note that although President Trump is a politician, participants' responses indicated that anger toward him loaded with Factor 2 rather than with Factor 1, which included other politicians (e.g., governors, former President Biden). This distinction is data-driven and reflects the empirical factor structure rather than a theoretical assumption. A follow-up reliability analysis found factor 2 to have good internal consistency ($\alpha = .86$).

Anger Factor 3 accounted for 6.93% of the variance and was related to **general anger towards the virus** in terms of its social and health effects on people. Lastly, Anger Factor 4 explained 5.49% of the variance and was related to the **vaccine envy** expressed towards individuals who got access to vaccines before themselves. The factor loadings after Varimax rotation are presented in Table 3.

Results from descriptive statistics reveal that the highest levels of anger were directed at misinformation spreaders ($M = 73.21$, $SD = 32.15$) and President Trump's handling of COVID-19 ($M = 60.64$, $SD = 37.99$) (Table 3). In addition, when

Table 3. Descriptive Statistics and Factor Loadings for Exploratory Factor Analysis of COVID-Related Anger.

| Targets of Anger | M (SD) | Anger Factor 1 | Anger Factor 2 | Anger Factor 3 | Anger Factor 4 |
|---|---|---|---|---|---|
| Public health officials | 36.08 (33.90) | **.831** | −.195 | .133 | .026 |
| World health organization (WHO) | 34.95 (36.45) | **.811** | −.327 | .159 | .059 |
| Pharmaceutical companies | 37.41 (35.19) | **.805** | −.251 | .114 | .050 |
| Medical institutions or Healthcare system | 27.62 (33.33) | **.783** | −.251 | .096 | .135 |
| Politicians or political parties | 56.30 (30.34) | **.783** | .337 | .015 | −.094 |
| Former President Biden's handling of COVID-19 | 34.44 (34.05) | **.759** | −.219 | .174 | .059 |
| Governor and their policies | 38.84 (35.22) | **.733** | .202 | −.011 | .027 |
| Vaccine and vaccine mandate | 28.94 (36.07) | **.540** | −.592 | .295 | .070 |
| Wearing masks and social distancing | 22.22 (31.77) | **.516** | −.516 | .230 | .239 |
| Anti-vaxxers or vaccine-hesitant individuals | 51.33 (39.54) | −.166 | **.845** | −.018 | .141 |
| President Trump's handling of COVID-19 | 60.64 (37.99) | −.001 | **.834** | −.083 | −.085 |
| Individuals not adhering to mask-wearing and social distancing guidelines | 56.81 (35.52) | −.180 | **.817** | .164 | .106 |
| Misinformation spreaders of COVID-19 | 73.21 (32.15) | .004 | **.769** | .238 | −.005 |
| COVID-19 virus and its negative social and health effects on people | 56.94 (31.86) | .278 | .181 | **.893** | .015 |
| Jealousy or envy experienced when others are given the opportunity to receive a COVID-19 vaccine | 8.43 (18.89) | .109 | .082 | .015 | **.972** |
| Eigenvalues | | 6.15 | 3.04 | 1.04 | 0.82 |
| % of Variance | | 41.01 | 20.23 | 6.93 | 5.49 |

*Note.* Varimax rotation was used. Loadings in bold text are included in each factor (>.50); $N = 1002$; 3 cases were excluded.

aggregated into four factors, the highest level of anger was observed in Anger Factor 2, which included anger towards anti-vaxxers or vaccine-hesitant individuals, President Trump, individuals not adhering to safety guidelines, and misinformation spreaders ($M = 60.54$, $SD = 30.60$) (Table 4).

We employed PROCESS (Model 4; [66]) to estimate parallel indirect effects of four targets of anger on future vaccination behavior intention mediated through traditional and social media news use. We conducted four separate analyses with each target of anger as the independent variable. Across all models, age, sex, race, education level, and political ideology were included as covariates. We also included the other targets of anger as covariates.

As presented in Fig 2, we first assessed whether Anger Factor 1 was associated with future vaccination intention through traditional and social media news use. Results from the total effect model revealed that this was negatively

Table 4. Zero Order Correlation between Key Variables with Descriptive Statistics (N = 1,005).

| | M | SD | 1 | 2 | 3 | 4 | 5 | 6 | 7 |
|---|---|---|---|---|---|---|---|---|---|
| 1. Anger Factor 1 [a] | 35.17 | 26.36 | – | | | | | | |
| 2. Anger Factor 2 [a] | 60.54 | 30.60 | −.33*** | – | | | | | |
| 3. Anger Factor 3 [a] | 56.96 | 31.84 | .36*** | .17*** | – | | | | |
| 4. Anger Factor 4 [a] | 8.41 | 18.88 | .16*** | .09** | .10*** | – | | | |
| 5. Future Vaccination Intention[b] | 3.69 | 1.48 | −.59*** | .65*** | −.08* | .04 | – | | |
| 6. Traditional Media News Use[c] | 3.16 | 1.86 | −.19*** | .19*** | .03 | .10** | .21*** | – | |
| 7. Social Media News Use[c] | 2.45 | 1.44 | .13*** | −.03 | .17*** | .16*** | −.06 | −.00[d] | – |

*Note.* [a]Scale ranged from 1 to 100; [b]Scale ranged from 1 to 5; [c]Scale ranged from 1 to 7; [d]-.001.

* $p < .05$, ** $p < .01$, *** $p < .001$.

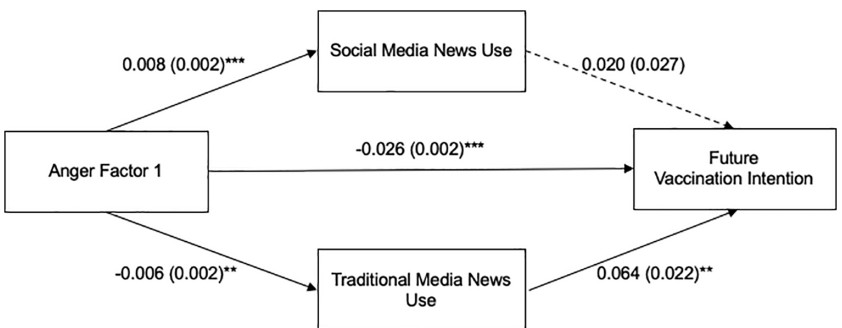

**Fig 2. Parallel Mediation Model of the Relationship between Anger Factor 1 and Future Vaccination Intention through Social Media and Traditional Media News Use.** Note. Unstandardized B was used to indicate path coefficients, with standard errors in parentheses. Paths represent associations, not causal relationships. Age, sex, race, education level, and political ideology were included in the model as covariates. **$p < .01$, ***$p < .001$.

associated with future vaccination intention, $b = -0.026$, $SE = 0.002$, $t = -17.78$, $p < .001$. The total effect model explained 44.50% of the variance in future vaccination intention, $R^2 = 0.445$, $F(10, 986) = 79.059$, $p < .001$. Anger Factor 1 was positively related to social media news use, $b = 0.008$, $SE = 0.002$, $t = 4.34$, $p < .001$, and negatively associated with traditional media news use, $b = -0.006$, $SE = 0.002$, $t = -2.94$, $p = .003$. Anger Factor 1 was negatively associated with future vaccination intention, $b = -0.026$, $SE = 0.002$, $t = -17.40$, $p < .001$. However, social media news use was not associated with future vaccination intention, $b = 0.020$, $SE = 0.027$, $t = 0.77$, $p = .444$, while traditional media news use was positively associated with future vaccination intention, $b = 0.064$, $SE = 0.022$, $t = 2.99$, $p = .003$.

The indirect effect of Anger Factor 1 on future vaccination intention via social media news use was not significant ($b = 0.0002$, Boot $SE = 0.0002$, 95% CI = [−0.0003, 0.001]). However, the indirect effect via traditional media news use was significant ($b = -0.0004$, Boot $SE = 0.0002$, 95% CI = [−0.001, −0.0001]). This indicates that traditional media news use mediates the relationship between Anger Factor 1 and future vaccination intention, where individuals angry toward health officials and agencies, politicians, and COVID-19 health policies were less likely to use traditional media news, which was associated with lower future vaccination intentions. For Anger Factor 1, adding social media and traditional media news use as mediators in the model yielded a statistically significant amount of additional variance in future vaccination intention ($R^2 = 0.451$, $\Delta R^2 = 0.006$, $p < .001$).

Next, we assessed the same parallel mediation model with Anger Factor 2, which includes anger targeted at anti-vaxxers, individuals not adhering to COVID-19 policies and guidelines, misinformation spreaders, and the President Trump, as the independent variable (Fig 3). Based on the total effect model, it was positively associated with future vaccination intention, $b = 0.026$, $SE = 0.001$, $t = 18.38$, $p < .001$. The total effect model explained 45.40% of the variance in future vaccination intention $R^2 = 0.454$, $F(10, 986) = 81.996$, $p < .001$. The results indicate that Anger Factor 2 was not significantly related to social media news use, $b = -0.001$, $SE = 0.002$, $t = -0.35$, $p = .725$, while it was positively related to traditional media news use, $b = 0.009$, $SE = 0.002$, $t = 4.22$, $p < .001$. Next, Anger Factor 2 was positively associated with future vaccination behavior, $b = 0.026$, $SE = 0.001$, $t = 17.91$, $p < .001$. Social media news use was not associated with future vaccination intention, $b = -0.036$, $SE = 0.026$, $t = -1.38$, $p = .167$, whereas greater traditional media news use was associated with greater future vaccination intention, $b = 0.054$, $SE = 0.022$, $t = 2.50$, $p = .013$.

For Anger Factor 2, the indirect effects on future vaccination intention via social media news use was not significant ($b = 0.0000$, Boot $SE = 0.0001$, 95% CI = [−0.0001, 0.0002]). However, the indirect effect via traditional media news use was significant ($b = 0.001$, Boot $SE = 0.0002$, 95% CI = [0.0001, 0.001]). This indicates that traditional media news use mediates the relationship between Anger Factor 2 and future vaccination intention, where individuals expressing anger towards anti-vaxxers, individuals not adhering to COVID-19 policies and guidelines, misinformation spreaders, and the

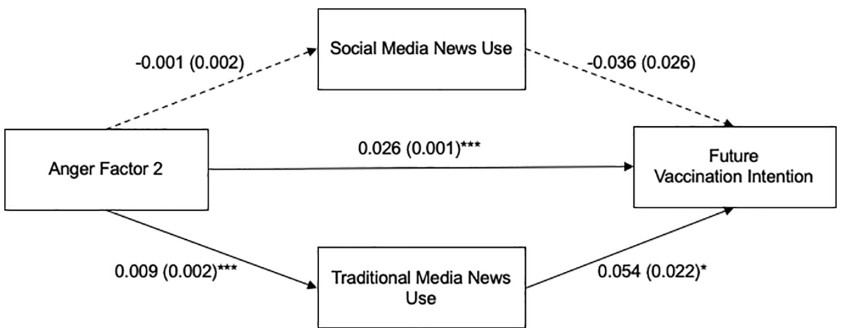

**Fig 3. Parallel Mediation Model of the Relationship between Anger Factor 2 and Future Vaccination Intention through Social Media and Traditional Media News Use.** Note. Unstandardized B was used to indicate path coefficients, with standard errors in parentheses. Paths represent associations, not causal relationships. Age, sex, race, education level, and political ideology were included in the model as covariates. *$p < .05$, ***$p < .001$.

President Trump were more likely to use traditional news media, which was associated with higher vaccination intentions. For Anger Factor 2, adding social media and traditional media news use as mediators in the model yielded a statistically significant amount of additional variance in future vaccination intention ($R2 = 0.452$, $\Delta R2 = 0.004$, $p = .024$).

Then, we assessed whether Anger Factor 3, general anger towards the virus, was associated with future vaccination intention via social media news use and traditional media news use (Fig 4). Based on the total effect model, Anger Factor 3 was negatively associated with future vaccination intention, $b = -0.003$, $SE = 0.001$, $t = -2.274$, $p = .023$. The total effect model explained 27.10% of the variance in future vaccination intention $R^2 = 0.271$, $F (10, 986) = 36.622$, $p < .001$. Results indicate that Anger Factor 3 was significantly related to social media news use, $b = 0.006$, $SE = 0.001$, $t = 4.709$, $p < .001$, and traditional media news use, $b = 0.004$, $SE = 0.002$, $t = 2.128$, $p = .034$. Next, Anger Factor 3 was positively associated with future vaccination intention, $b = -0.003$, $SE = 0.001$, $t = -2.378$, $p = .018$. Social media news use was not significantly associated with future vaccination intention, $b = -0.038$, $SE = 0.030$, $t = -1.260$, $p = 0.208$, while traditional media news use was significantly associated with future vaccination intention $b = 0.109$, $SE = 0.025$, $t = 4.443$, $p < .001$.

The indirect effect via social media news use was not significant ($b = -0.0002$, Boot $SE = 0.0002$, 95% CI = [−0.001, 0.0001]). In contrast, the indirect effect via traditional media news use was significant ($b = 0.0004$, Boot $SE = 0.0002$, 95% CI = [0.00, 0.001]). This indicates that traditional media news use mediates the relationship between general anger towards COVID-19 virus and future vaccination intention. For Anger Factor 3, adding social media and traditional media

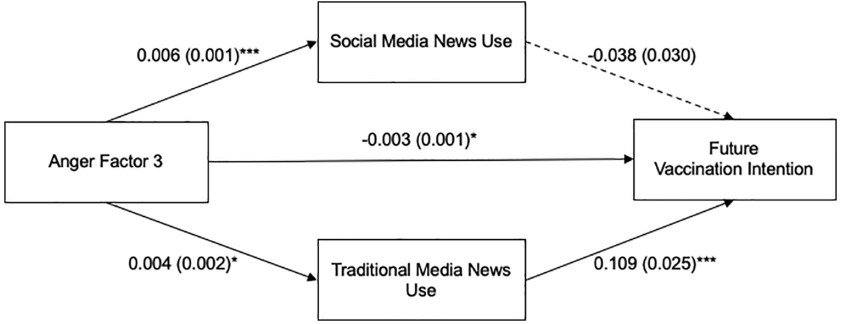

**Fig 4. Parallel Mediation Model of the Relationship between Anger Factor 3 and Future Vaccination Intention through Social Media and Traditional Media News Use.** Note. Unstandardized B was used to indicate path coefficients, with standard errors in parentheses. Paths represent associations, not causal relationships. Age, sex, race, education level, and political ideology were included in the model as covariates. *$p < .05$, ***$p < .001$.

news use as mediators in the model yielded a statistically significant amount of additional variance in future vaccination intention ($R2 = 0.286$, $\Delta R2 = 0.015$, $p < .001$).

Lastly, we examined whether Anger Factor 4 was related to future vaccination intention through social media news use and traditional media news use (Fig 5). Results from the total effect model revealed that Anger Factor 4 was positively associated with future vaccination intention, $b = 0.004$, $SE = 0.002$, $t = 2.043$, $p = .041$. The total effect model explained 27% of the variance in future vaccination intention, $R^2 = 0.270$, $F (10, 985) = 36.391$, $p < .001$. Anger Factor 4 was positively associated with social media news use, $b = 0.013$, $SE = 0.002$, $t = 5.782$, $p < .001$, and with traditional media news use, $b = 0.007$, $SE = 0.003$, $t = 2.559$, $p = .011$. Next, Anger Factor 4 was significantly associated with future vaccination behavior, $b = 0.004$, $SE = 0.002$, $t = 2.041$, $p = 0.042$. Social media news use was not significantly associated with future vaccination intention, $b = −0.059$, $SE = 0.031$, $t = −1.947$, $p = .052$. On the other hand, traditional media news use was positively associated with future vaccination intention, $b = 0.102$, $SE = 0.025$, $t = 4.179$, $p < .001$.

The indirect effect through social media news use was not significant ($b = −0.001$, Boot $SE = 0.0004$, 95% CI = [−0.002, 0.000]). However, the indirect effect via traditional media news use was significant ($b = 0.001$, Boot $SE = 0.0004$, 95% CI = [0.0001, 0.002]). This indicates that traditional media news use mediates the relationship between vaccine envy and future vaccination intention. For Anger Factor 4, adding social media and traditional media news use as mediators in the model yielded a statistically significant amount of additional variance in future vaccination intention ($R2 = 0.284$, $\Delta R2 = 0.014$, $p < .001$).

## Study 3 Discussion

These findings suggest that the relationship between anger, media news use, and future vaccination intentions depends on the target of anger. For example, anger directed at health officials and agencies, politicians, and preventive health behaviors (Anger Factor 1) was negatively associated with traditional media use, which in turn was linked to lower vaccination intention, whereas anger directed at anti-vaxxers, individuals not adhering to COVID-19 policies and guidelines, misinformation spreaders, and President Trump (Anger Factor 2) was positively associated with both traditional media news use and vaccination intention. These findings highlight that anger, depending on its target, can lead to different patterns of traditional media news consumption, which in turn shape vaccination intention during a health crisis.

## General discussion

### Principle findings and theoretical implications

Anger is an understudied and potentially misunderstood emotion that warrants greater scholarly attention [22]. Understanding the target of public anger is crucial, as anger can influence the actions people take [7]. Across three studies,

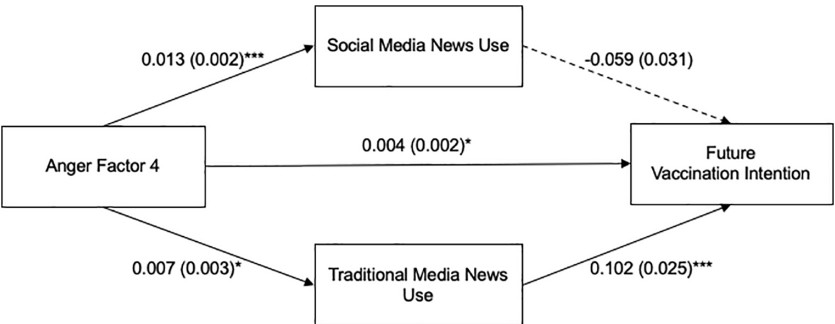

**Fig 5. Parallel Mediation Model of the Relationship between Anger Factor 4 and Future Vaccination Intention through Social Media and Traditional Media News Use.** Note. Unstandardized B was used to indicate path coefficients, with standard errors in parentheses. Paths represent associations, not causal relationships. Age, sex, race, education level, and political ideology were included in the model as covariates. *$p < .05$, ***$p < .001$.

we examined how anger and news media use were related to pandemic vaccination intentions and the way anger was expressed on social media in COVID-19 discourse. In Study 1, we found that anger towards COVID-19 was associated with increased social and traditional media news use, which was then related to stronger intentions to get vaccinated for COVID-19. In Study 2, drawing on appraisal theories of emotions (e.g., [19]), we identified 15 targets of anger from naturalistic anger expressions in social media discourse about COVID-19. Study 3 replicated and extended Study 1 and 2 findings by examining how media news consumption may connect the relationship between anger directed at various targets and intentions to get vaccinated in the event of a future pandemic similar to COVID-19. Our findings from Studies 1 and 3 consistently highlight that different types of anger were associated with increased traditional media news use, which in turn was related to greater intent to obtain the COVID-19 vaccine and a hypothetical future vaccine in the case of another pandemic. One exception to this was found in Study 3, which revealed that anger towards health officials and agencies, politicians, and health policies was linked to less traditional media news consumption and was associated with lower intentions to vaccinate. This finding suggests that individuals who are angry about governments and public health policies may avoid traditional media because the majority of these outlets often support public health measures, as traditional media more frequently publish information from high quality sources [50,82].

With regard to social media news use, the findings of Study 1 and 3 differed. Specifically, social media news use mediated the relationship between general anger towards COVID-19 virus and intention to vaccinate, such that anger was positively associated with social media use and intention to vaccinate in Study 1. The mediating effect of social media news use, however, disappeared in Study 3 across all of the targets of anger. Only the Anger Factor 4 model (see Fig 5) yielded a marginal mediating effect of social media news use.

These findings may signal that the role of social media as a source of information may have changed after the pandemic. First, during the early stages of the pandemic, social media was a good alternative to seek and share information. To cope with the anxiety related to the pandemic, people increasingly turned to social media platforms for news [45,83]. However, the unprecedented reliance on social media for information consumption during the pandemic gave rise to an "infodemic," which was characterized by the mass availability of information online [52,84]. This "infodemic" could have prompted social media fatigue and resulted in information overload [85]. In turn, social media usage patterns may have changed as information overload exhausted people, resulting in reduced usage [86,87]. Given that our data in Study 3 were collected in the post-emergency declaration climate, it is possible that social media news use did not have the same impact on vaccination intentions as during the COVID-19 pandemic.

Although we did not measure participants' exposure to misinformation in this study, it is possible that increased anger towards COVID-19-related issues led to information seeking through media [59] and at the same time increased one's exposure to and engagement with misinformation [40,88,89]. Previous research also highlights that social media as a channel for news can be associated with anti-vaccine attitudes [14,90]. In other words, although anger is associated with increased consumption of social media news, being exposed to COVID-19 misinformation in online settings may backfire and even decrease one's intention to receive vaccination. In this instance, simply correcting misinformation may not be effective. Prior research has demonstrated that even after misinformation is corrected, it can continue to influence individuals' beliefs—a phenomenon known as the continued influence effect [91].

## Methodological implications

Leveraging a human coded dataset with a fine-tuned GPT model in analyzing large-scale social media data presents several practical implications for social science research. First, the AI-driven approach enhances the efficiency of data analysis. By automating the classification process, the GPT model reduces the time and effort required to process and interpret vast amounts of data. For instance, prior research has demonstrated that the zero-shot accuracy of ChatGPT outperformed crowd workers for text-annotation tasks across multiple datasets [92]. Similarly, others found higher accuracy, reliability, and equal or lower bias of classification performed by ChatGPT-4 when compared to human classifiers in

classifying twitter messages [93]. This highlights the potential of leveraging large language models (LLMs) to benefit social science research by obtaining efficient, accurate, and reliable results.

In addition, the methodological merit of using *instruction tuning* and *supervised fine-tuning* [76,77] using LLMs is significant for health communication. This approach allows for quick adaptation to classify nuanced vaccine-related public sentiments efficiently, without the need for extensive retraining. This reduces time and computational cost while maintaining accuracy, illustrating a scalable approach for analyzing large-scale communication data. Beyond the current study, this methodological framework could be extended to other contexts of mediated health communication, offering researchers a replicable strategy for monitoring emerging health issues and comparing different sentiment across domains.

## Practical implications

Anger is not necessarily an emotion that results in negative consequences and can motivate individuals to take action. For example, past work has found tweets expressing anger were associated with increased online engagement [94,95]. Because anger is linked to action tendencies [7], it is important for health communication scholars to examine how anger manifests in public discourse and how it may shape responses to health issues. Recent work suggests that anger appeals may be associated with greater health-related policy support [96]. These findings point to the importance of monitoring and understanding anger as it arises in mediated communication, and of considering strategies to address it constructively, such as by providing accurate information or encouraging preventative health behaviors.

Anger was associated with information seeking through both traditional and social news media. For all four anger factors in Study 3, we found that traditional news use consistently mediated the relationship between anger and vaccination intentions. This illustrates the importance of incorporating traditional news media in our media diets. Although misinformation exists all over the media environment, practitioners may consider working with traditional news sources of health information, rather than social media sources, especially due to the lack of regulation for misleading, inaccurate, and hyperpartisan news on social media. Our findings suggest that traditional media use was associated with vaccination intention, demonstrating traditional news media's potential positive impacts for health information and decision making. This also highlights the potential for strategic collaborations between traditional news media and health officials. Especially with the reach of traditional news media, partnerships with health professionals on television may aid in providing accurate, trustworthy health information to its audiences. Future studies could experimentally examine how audience trust influences the effectiveness of collaborations between traditional media and health officials. Because both sources are subject to declining levels of public trust [97,98], it is important to test whether pairing these sources leads to stronger persuasive effects through enhanced credibility, or weaker effects if distrust toward one source carries over to the other. Further, media messages highlighting vaccine-supportive policies strengthen vaccination behaviors [99], emphasizing the importance of incorporating this messaging in news media.

In addition to promoting traditional news media sources for health information, a major focus of practitioners should be to push accurate, theoretically-driven messages on their social media accounts to promote vaccination. If anger is driving individuals to turn to online news sources, practitioners need to ensure that individuals have increased access to public health promotion messaging, while limiting the exposure to messages against vaccination and other preventive measures. By making pro-vaccination beliefs more salient, or highlighting new beliefs, practitioners can still encourage vaccination despite the presence of misleading or inaccurate messages on social media [100,101]. In addition, practitioners may aim to focus on policy efforts to improve social media companies' regulation of their platforms and increase access to reliable health information. For instance, vaccine-related misinformation could be corrected prior to its spread through prebunking, which has been shown to be effective in reducing misinformation engagement [102], although evidence is mixed [103]. There is also evidence that accuracy nudges can increase deliberation and reduce misinformation engagement [104]. Lastly, acknowledging and affirming the emotions of the public may increase receptivity to misinformation correction [105,106].

## Limitations

We employed two surveys and a large-scale social media analysis covering the COVID-19 pandemic from the beginning stages to current day; however, this study is not without limitations. First, we ran mediation path analyses to examine the role of anger on news media use and vaccination intentions, but cannot claim causality from our studies. Future work may employ experiments to determine whether anger has a causal impact on these outcome variables. Second, we employed single-item scales to measure 15 distinct types of anger, aiming to balance measurement flexibility with minimal participant burden. While we recognize potential concerns regarding the validity of single-item scales, the structural similarity among the anger types suggested that using multiple-item scales potentially lead to participant fatigue, reduced response rates, and less accurate data. Therefore, single-item scales were deemed most appropriate to minimize respondent overload.

Third, we utilized LIWC [70] to measure expressed anger within tweets related to COVID-19. While this dictionary-based software has been well-validated, other computational tools may be employed to determine the level of anger expressed on social media. One such tool we used is the GPT model for coding, which introduces potential biases from the training data [107]. However, we mitigated this by fine-tuning the model with our own dataset, ensuring more relevant and accurate classifications. Concerns about accuracy and misclassification were addressed by implementing a low temperature [108], or randomness, parameter during fine-tuning, which enhanced the precision of the model's outputs. Another limitation is related to the use of a predefined codebook to classify targets of COVID-19 related anger. While the categories were informed by prior research and refined through close examination of our dataset, this approach may not capture all possible nuances in how anger was expressed. Some subtle or less frequent targets may have been overlooked, and future work employing unsupervised classification methods may uncover additional emergent categories. Twitter/X has a set of affordances that make it different from social media and its users are not a representative sample of the U.S. population [109], none-the-less it still provided insights on the targets of anger related to COVID-19.

Finally, it must be noted that our sample in Study 3 was significantly skewed towards liberals, which is not truly representative of the U.S. population and could have impacted the findings. One explanation for this is that our sample in Study 3 was chosen to be representative of the population based on demographics such as age, sex, and ethnicity but not political ideology. In addition, we relied on Prolific, which is a nonprobability sample and is prone to self-selection [110]. Still, to minimize bias due to affiliation, we controlled for political ideology in all our analyses in both Studies 1 and 3.

## Conclusion

Anger is an understudied and complex emotion that deserves more scholarly attention due to its ability to drive behavior. In the current paper, we employed a computational approach to reveal how anger manifested towards different targets during COVID-19. Additionally, utilizing survey data from the early stage of COVID-19 pandemic and post-emergency declaration, we found that, depending on the blamable target, anger was associated with intentions to obtain the COVID-19 vaccine and a vaccine for a hypothetical future pandemic through media news use, especially traditional media news use. We contend that anger is a nuanced emotion, and under some conditions, it can enhance preventive health behaviors during public health crises.

## Supporting information

**S1 File. Study 2 GPT Prompt.**
(DOCX)

**S2 File. Study 2 GPT Prompt: Additional Classification for Others Category.**
(DOCX)



## Author contributions

**Conceptualization:** Yoo Jung Oh, Muhammad Ehab Rasul, Jong In Lim, Christopher Calabrese.

**Data curation:** Yoo Jung Oh, Muhammad Ehab Rasul, Jong In Lim, Emily McKinley, Hannah Stevens, Monique Mitchell Turner, Maria Knight Lapinski, Tai-Quan Peng.

**Formal analysis:** Yoo Jung Oh, Muhammad Ehab Rasul, Jong In Lim, Emily McKinley.

**Funding acquisition:** Monique Mitchell Turner, Maria Knight Lapinski, Tai-Quan Peng.

**Investigation:** Yoo Jung Oh, Muhammad Ehab Rasul, Christopher Calabrese.

**Methodology:** Yoo Jung Oh, Muhammad Ehab Rasul, Jong In Lim, Christopher Calabrese, Emily McKinley, Hannah Stevens.

**Writing – original draft:** Yoo Jung Oh, Muhammad Ehab Rasul, Jong In Lim, Christopher Calabrese, Emily McKinley, Hannah Stevens.

**Writing – review & editing:** Yoo Jung Oh, Muhammad Ehab Rasul, Jong In Lim, Christopher Calabrese, Emily McKinley, Hannah Stevens, Monique Mitchell Turner, Maria Knight Lapinski, Tai-Quan Peng.

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
