## [Decision Letter · Decision Letter 0]

27 Aug 2025

Dear Dr. Oh,

Thank you for submitting your manuscript to PLOS ONE. After careful consideration, we feel that it has merit but does not fully meet PLOS ONE’s publication criteria as it currently stands. Therefore, we invite you to submit a revised version of the manuscript that addresses the points raised during the review process.

We look forward to receiving your revised manuscript.

Kind regards,

Ruobing Li, Ph.D.

Academic Editor

PLOS ONE

Journal Requirements:

Reviewers' comments:

Reviewer's Responses to Questions

**Comments to the Author**

1. Is the manuscript technically sound, and do the data support the conclusions?

Reviewer #1: Yes

Reviewer #2: Yes

2. Has the statistical analysis been performed appropriately and rigorously?

Reviewer #1: Yes

Reviewer #2: Yes

3. Have the authors made all data underlying the findings in their manuscript fully available?

Reviewer #1: Yes

Reviewer #2: No

4. Is the manuscript presented in an intelligible fashion and written in standard English?

Reviewer #1: Yes

Reviewer #2: Yes

Reviewer #1: It is my pleasure to review the manuscript entitled Targeting Anger for COVID-19 Prevention: The Motivating Role of Anger on Media Use and Vaccination Intention. This manuscript addresses a valuable topic, uses solid methods, and is clearly written. At this stage, however, I have some concerns that are listed below.

1. The research gap presented in this study is a bit vague. In page 3, line 60, the authors stated that “there is minimal scholarly attention investigating how anger and media use collectively are associated with COVID-19 vaccine behavior”. Nonetheless, I found that there are several papers addressing this issue, e.g., Ali et al., (2022); Featherstone et al., (2020); and Silar-Taut and Mican (2022). The authors may consider making a stronger case about the research gap.

Ali, K., Li, C., Zain-ul-abdin, K., & Muqtadir, S. A. (2022). The effects of emotions, individual attitudes towards vaccination, and social endorsements on perceived fake news credibility and sharing motivations. Computers in Human Behavior, 134, Article 107307. https://doi.org/10.1016/j.chb.2022.107307

Featherstone, J. D., & Zhang, J. (2020). Feeling angry: The effects of vaccine misinformation and refutational messages on negative emotions and vaccination attitude. Journal of Health Communication, 25(9), 692–702. https://doi.org/10.1080/10810730.2020.1838671

Sitar-Taut, D.-A., & Mican, D. (2022). Social media exposure assessment: Influence on attitudes toward generic vaccination during the COVID-19 pandemic. Online Information Review, 47(1), 138–161. https://doi.org/10.1108/OIR-11-2021-0621

2. Page 5, line 102: The author stated that only anger can be oriented to a specific individual or entity. I don’t quite agree with the statement. For instance, fear can be directed towards a specific entity (e.g., fear towards criminals; fear towards a virus). As such, I think anger, though being a unique emotion, might not be unique in this way. The authors may clarify more about the uniqueness of anger or refrain from this and similar statements in other parts of this manuscript.

3. Page 6, lines 120-126. The argument is a bit odd for me. I concur that anger leads to increased information seeking and processing. Nonetheless, this can’t lead to the argument that investigating different types of media is important. The authors may make a stronger case about this.

4. In study 1, why social media use was measured by the frequencies while traditional media use by the number of minutes?

5. I’m a bit confused by the aim and actual methods of study 2. If the authors intend to identify the target of anger, why is there a pre-decided codebook rather than identifying the targets of anger using a bottom-up method (classifying targets based on the data using other computational methods)?

6. Page 22, line 458: A minor issue: Trump is now president.

7. In study 3, factors of anger are named anger factor 1, 2, and 3. I think more meaningful names will make the study easier to understand for readers.

8. Why are the indirect associations between the three factors of anger tested in three separate models rather than one model with three independent variables?

Reviewer #2: This is a well-conceived and carefully executed project. It tackles an important issue in research about emotions – what the targets of a particular emotion are and how this affects attitudinal and behavioral outcomes. A fair amount of research has been conducted to document the experience of anger during the COVID pandemic. People may have good intuition in terms of the targets and nature of this emotional experience. However, empirical studies are extremely limited. This is one of the first studies that employed a three-study design to investigate this issue. The authors are apparently well-versed in relevant literature and methodologically versatile. The manuscript is clearly organized and accessible, and the research questions are thoughtfully derived from the reviewed literature. The findings contribute meaningfully to the scholarship on the impact of anger, especially in communication and public health.

I have only one question regarding the social media data in the second study. There is a category of “Others (unspecified)” when coding the tweets for anger targets. If these are tweets with anger expressed but no obvious targets, I think it would be fine. However, there seems to be a subcategory in this group that did not express any anger, as noted on p.18, line 357. Indeed, the example provided in that paragraph does not seem to contain any reference of anger. This contradicts to what is mentioned in the codebook (in the supporting document) about this “Others” category where it says that “This category encompasses tweets where anger is expressed in relation to COVID-19 but does not specifically target any of the entities or issues outlined in categories 1-14.” Given that this category is disproportionately large (48% of the total tweets), and how large this subgroup (no anger expressed) is, I’d like to hear the authors’ thoughts on this. How much confidence can we put in the fine-tuned GPT model for identifying the anger tweets?

**Do you want your identity to be public for this peer review?** For information about this choice, including consent withdrawal, please see our Privacy Policy

Reviewer #1: No

Reviewer #2: No

---

## [Author Response · Author response to Decision Letter 1]

28 Oct 2025

Response to Reviewers’ Comments

Reviewer #1

It is my pleasure to review the manuscript entitled Targeting Anger for COVID-19 Prevention: The Motivating Role of Anger on Media Use and Vaccination Intention. This manuscript addresses a valuable topic, uses solid methods, and is clearly written. At this stage, however, I have some concerns that are listed below.

Response: Thank you for your favorable comments regarding the manuscript’s main focus, as well as our methodology and writing. We believe we have addressed your concerns in the following responses below.

1. The research gap presented in this study is a bit vague. In page 3, line 60, the authors stated that “there is minimal scholarly attention investigating how anger and media use collectively are associated with COVID-19 vaccine behavior”. Nonetheless, I found that there are several papers addressing this issue, e.g., Ali et al., (2022); Featherstone et al., (2020); and Silar-Taut and Mican (2022). The authors may consider making a stronger case about the research gap.

Ali, K., Li, C., Zain-ul-abdin, K., & Muqtadir, S. A. (2022). The effects of emotions, individual attitudes towards vaccination, and social endorsements on perceived fake news credibility and sharing motivations. Computers in Human Behavior, 134, Article 107307. https://doi.org/10.1016/j.chb.2022.107307

Featherstone, J. D., & Zhang, J. (2020). Feeling angry: The effects of vaccine misinformation and refutational messages on negative emotions and vaccination attitude. Journal of Health Communication, 25(9), 692–702. https://doi.org/10.1080/10810730.2020.1838671

Sitar-Taut, D.-A., & Mican, D. (2022). Social media exposure assessment: Influence on attitudes toward generic vaccination during the COVID-19 pandemic. Online Information Review, 47(1), 138–161. https://doi.org/10.1108/OIR-11-2021-0621

Response: Thank you for bringing this point to our attention. Our set of studies focus on the motivating role of anger on news media use and vaccination intentions. Because intentions are a more direct predictor of behavior than attitudes (Albarracin et al., 2001), our studies take a step further from prior work examining general attitudes toward vaccines to examine how anger itself may more likely motivate COVID-19 vaccination intentions.  

The referenced studies do however provide a good picture for the relationship between message effects and anger. Featherstone and Zhang (2020) examined the effects of refutational messages for addressing vaccine misinformation on general vaccine attitudes, where the refutational messages increase vaccine attitudes compared to the control through reduced anger in response to the messages. Ali et al. (2022) examine the role of anger and its interaction with pre-existing attitudes, finding that participants with neutral attitudes toward vaccines who were anger-induced were more motivated to have higher self-expression sharing motivations compared to those who were fear-induced. Further, Sitar-Taut and Mican (2022) find that media exposure related to COVID-19 was associated with anger toward COVID-19, which was then associated with reduced preventive behaviors and attitudes (related to social distancing, hand-washing and masking), but not vaccine attitudes. While these studies examine anger, they do not focus on anger as a driver for motivating news media use and COVID-19 vaccination, rather, they focus on vaccine attitudes or message sharing intentions. In our studies, we focus on anger’s impact alone and its behavioral tendency to prompt action through traditional/social media use and vaccination intentions. Our studies also tease out the different targets of anger related to COVID-19, which may also explain the null results for the relationship between vaccine attitudes and anger in Sitar-Taut and Mican’s (2022) study.

In addition, we do recognize that there is research on anger and COVID-19. And we agree that our argument can be further specified and have added the following to the manuscript on Pages 3-4:

“Although there is research focused on anger and COVID-19 (e.g., [11, 12]), and social media use and COVID-19 vaccine hesitancy [13, 14], there is minimal scholarly attention investigating how anger and media use collectively are associated with COVID-19 vaccine behavior despite the need for improving vaccine uptake. Prior work has found that refutational messages may influence general vaccination attitudes through reduced anger [15], and that anger towards COVID-19 is associated with COVID-19 media exposure, but not general vaccine attitudes [16]. However, less is known about anger as a potential driving force for news media use, as well as for COVID-19 vaccination intentions, since intentions are a more direct, specific predictor of behavior than general attitudes [17, 18].”

References:

Featherstone, J. D., & Zhang, J. (2020). Feeling angry: The effects of vaccine misinformation and refutational messages on negative emotions and vaccination attitude. Journal of Health Communication, 25(9), 692-702.

Sitar-Taut, D. A., & Mican, D. (2023). Social media exposure assessment: influence on attitudes toward generic vaccination during the COVID-19 pandemic. Online Information Review, 47(1), 138-161.

Albarracin, D., Johnson, B. T., Fishbein, M., & Muellerleile, P. A. (2001). Theories of reasoned action and planned behavior as models of condom use: a meta-analysis. Psychological bulletin, 127(1), 142.

Kim, M.-S., & Hunter, J. E. (1993). Relationships among attitudes, behavioral intentions, and behavior: A meta-analysis of past research: II. Communication Research, 20(3), 331–364. https://doi.org/10.1177/009365093020003001

2. Page 5, line 102: The author stated that only anger can be oriented to a specific individual or entity. I don’t quite agree with the statement. For instance, fear can be directed towards a specific entity (e.g., fear towards criminals; fear towards a virus). As such, I think anger, though being a unique emotion, might not be unique in this way. The authors may clarify more about the uniqueness of anger or refrain from this and similar statements in other parts of this manuscript.

Response: We thank the reviewer for this insightful comment. We agree that our original wording may have overstated the uniqueness of anger in terms of being directed at a specific target. On Page 5, we emphasize that what distinguishes anger is not merely its directedness, but rather its attribution of blame and responsibility to a particular entity and its motivation of confrontational responses. 

“In theory, what distinguishes anger from other emotions is its inherent blame attribution. Anger involves a blamable target who is perceived to be responsible for obstructing one's personal goals [7, 19, 20]. Additionally, anger is considered an approach emotion (i.e., one that drives behavior, rather than an avoidance mechanism) because angry individuals are more prone to take action to cope with their obstructed goals [21, 22]. Anger is uniquely characterized by its tendency to attribute responsibility to specific agents and motivate responses to those blamed targets.” 

3. Page 6, lines 120-126. The argument is a bit odd for me. I concur that anger leads to increased information seeking and processing. Nonetheless, this can’t lead to the argument that investigating different types of media is important. The authors may make a stronger case about this.

Response: Thank you for this insightful comment. We have added additional arguments and now focus on how during health crises such as COVID-19, individuals are likely to turn to news media outlets for information and stay up to date. Specifically, we have added the following on pages 6-7 in the section titled “Media Use as the Mediator”:

“When people experience anger about health threats such as COVID-19, they are likely to engage in news media consumption both to identify who or what is to blame for the threat and to seek guidance [41]. Prior research has shown that times of crisis and uncertainty are accompanied by increases in news media consumption [42, 43]. However, not all media serve the same role. Traditional media outlets often provide live briefings from public health officials and leaders, positioning them as primary sources of information [44]. At the same time, in times of crisis, the preference for immediate news [43] makes social media an attractive source of information. During the COVID-19 pandemic, an unprecedented demand for information, combined with stay-at-home orders and an increase in remote work resulted in high reliance on social media for information [45]. Taken together, these patterns suggest that it is essential to understand not only whether anger drives information seeking, but also the types of media individuals turn to when experiencing anger toward COVID-19 [46-48]. Notably, two prominent information sources individuals sought out for COVID-19 are social and traditional news media [48, 49].” 

References

Bento, A. I., Nguyen, T., Wing, C., Lozano-Rojas, F., Ahn, Y.-Y., & Simon, K. (2020). Evidence from internet search data shows information-seeking responses to news of local COVID-19 cases. Proceedings of the National Academy of Sciences of the United States of America, 117(21), 11220–11222. https://doi.org/10.1073/pnas.200533511

Van Aelst, P., Toth, F., Castro, L., Štětka, V., Vreese, C. de, Aalberg, T., Cardenal, A. S., Corbu, N., Esser, F., Hopmann, D. N., Koc-Michalska, K., Matthes, J., Schemer, C., Sheafer, T., Splendore, S., Stanyer, J., Stępińska, A., Strömbäck, J., & Theocharis, Y. (2021). Does a crisis change news habits? A comparative study of the effects of COVID-19 on news media use in 17 European countries. Digital Journalism, 9(9), 1208–1238. https://doi.org/10.1080/21670811.2021.1943481

Westlund, O., & Ghersetti, M. (2015). Modelling News Media Use: Positing and Applying the GC/MC Model to the Analysis of Media Use in Everyday Life and Crisis Situations. Journalism Studies, 16(2), 133–151.

Newman, N.,. R. Fletcher, A. Schulz, S. Andi, and R. K. Nielsen. “Digital News Report 2020.” Reuter Institute for the Study of Journalism. 

Karami, A., Zhu, M., Goldschmidt, B., Boyajieff, H. R., & Najafabadi, M. M. (2021). COVID-19 vaccine and social media in the US: Exploring emotions and discussions on Twitter. Vaccines, 9(10), 1059

4. In study 1, why social media use was measured by the frequencies while traditional media use by the number of minutes?

Response: Thank you for this comment. In Study 1, social and traditional media use were measured differently based on the available measures in the dataset. We acknowledge that this limits direct comparability between media types. Importantly, our analyses do not rely on direct comparisons between social and traditional media use, but rather examine their separate associations with outcomes. To address this limitation, in Study 3 we employed consistent measures for both social and traditional media use, which allows for more direct comparability.

5. I’m a bit confused by the aim and actual methods of study 2. If the authors intend to identify the target of anger, why is there a pre-decided codebook rather than identifying the targets of anger using a bottom-up method (classifying targets based on the data using other computational methods)?

Response: Thank you for your comment. Our approach to developing the codebook was informed by both prior research and the data itself. Specifically, we drew on existing literature (e.g., vaccine envy, anger towards institutions; [26-28, 70]) and social media analyses that identified recurring targets of COVID-related anger (e.g., “vaccine envy,” pharmaceutical companies, government officials and policies). Using these as a starting point, we then examined our dataset and refined the categories into an initial codebook. As laid out on page 15, two researchers independently examined subsets of tweets to assess whether the preliminary categories adequately captured the expressions of anger, while also noting emergent targets that were not reflected in prior work. Through iterative discussion, we refined the codebook by merging overlapping categories, clarifying definitions, and adding new categories that were inductively derived from the data. Thus, the final codebook reflects both existing knowledge and empirical insights from the dataset, balancing a top-down and bottom-up approach.

“We analyzed a dataset of 5,730 anger-related tweets to identify specific targets of anger. Given that several targets of anger were identified in previous studies such as anger towards individuals who do not adhere to safety guidelines or are reluctant to receive vaccination, those who had vaccine opportunity ahead others (i.e., vaccine envy), government, and policies regarding COVID-19 (e.g., [27-29, 71]), we developed a preliminary codebook informed by prior literature. Then, in the manual review process, two researchers independently examined subsets of tweets to assess whether the preliminary categories adequately captured the expressions of anger, while also noting any emergent targets not reflected in prior work. Through iterative discussion, we refined the codebook by merging overlapping categories, clarifying definitions, and adding new categories that were inductively derived from the data.”

6. Page 22, line 458: A minor issue: Trump is now president.

Response: Thank you for pointing this out. We have revised the manuscript accordingly.

7. In study 3, factors of anger are named anger factor 1, 2, and 3. I think more meaningful names will make the study easier to understand for readers.

Response: Thank you for this thoughtful suggestion. We agree that meaningful labels can help readers interpret factors more easily. In this case, however, we felt it was more appropriate to retain the neutral labels (“Anger Factor 1,” “Anger Factor 2,” “Anger Factor 3,” and “Anger Factor 4”). This is because there is no clear theoretical basis for assigning specific labels, and because some factors include categories that do not align neatly under one conceptual theme (e.g., Factor 1 includes both “politicians” and “President Biden,” while Factor 2 includes “misinformation spreaders” and “President Trump”), assigning labels could risk oversimplification or misinterpretation. To maintain clarity and transparency, we believe it was appropriate to keep the neutral labels while presenting the component categories in detail, allowing readers to interpret the factors directly.

8. Why are the indirect associations between the three factors of anger tested in three separate models rather than one model with three independent variables?

Response: Thank you for raising this important point. We tested the indirect associations in separate PROCESS models because the macro is designed to estimate mediation for one focal predictor at a time. Our goal was to examine the unique indirect pathway of each anger factor. However, we included the other factors as covariates in all analyses. This approach allowed us to account for overlap while still focusing on the focal predictor’s effect. While a structural equation modeling framework could in principle test all three predictors simultaneously, we chose PROCESS for its suitability to our research questions and to maintain clarity and comparability across models.

Reviewer #2

This is a well-conceived and carefully executed project. It tackles an important issue in research about emotions – what the targets of a particular emotion are and how this affects attitudinal and behavioral outcomes. A fair amount of research has been conducted to document the experience of anger during the COVID pandemic. People may have good intuition in terms of the targets and nature of this emotional experience. However, empirical studies are extremely limited. This is one of the first studies that employed a three-study design to investigate this issue. The authors are apparently well-versed in relevant literature and methodologically versatile. The manuscript is clearly organized and accessible, and the research questions are thoughtfully derived from the reviewed literature. The findings contribute meaningfully to the scho

---

## [Decision Letter · Decision Letter 1]

19 Nov 2025

Targeting Anger for COVID-19 Prevention: The Motivating Role of Anger on Media Use and Vaccination Intention

PONE-D-25-00520R1

Dear Dr. Oh,

We’re pleased to inform you that your manuscript has been judged scientifically suitable for publication and will be formally accepted for publication once it meets all outstanding technical requirements.

Kind regards,

Ruobing Li, Ph.D.

Academic Editor

PLOS ONE

Additional Editor Comments (optional):

Reviewers' comments:

Reviewer's Responses to Questions

**Comments to the Author**

Reviewer #1: All comments have been addressed

Reviewer #2: All comments have been addressed

2. Is the manuscript technically sound, and do the data support the conclusions?

Reviewer #1: Yes

Reviewer #2: Yes

3. Has the statistical analysis been performed appropriately and rigorously?

Reviewer #1: Yes

Reviewer #2: Yes

4. Have the authors made all data underlying the findings in their manuscript fully available?

Reviewer #1: Yes

Reviewer #2: No

5. Is the manuscript presented in an intelligible fashion and written in standard English?

Reviewer #1: Yes

Reviewer #2: Yes

Reviewer #1: Thank you for authors' efforts of revise this manuscript. My comments are well addressed in this version.

Reviewer #2: All my comments have been addressed. I appreciate the authors' thoughtful responses. Best of luck to them for their future research.

**Do you want your identity to be public for this peer review?** For information about this choice, including consent withdrawal, please see our Privacy Policy

Reviewer #1: No

Reviewer #2: No

---

## [Editor Report · Acceptance letter]

PONE-D-25-00520R1

PLOS One

Dear Dr. Oh,

I'm pleased to inform you that your manuscript has been deemed suitable for publication in PLOS One. Congratulations! Your manuscript is now being handed over to our production team.

Kind regards,

on behalf of

Dr. Ruobing Li

Academic Editor

PLOS One